# Dkk2 promotes neural crest specification by activating Wnt/β-catenin signaling in a GSK3β independent manner

Arun Devotta[1], Chang-Soo Hong[1,2], Jean-Pierre Saint-Jeannet[1]*

[1]Department of Basic Science and Craniofacial Biology, College of Dentistry, New York University, New York, United States; [2]Department of Biological Sciences, Daegu University, Gyeongsan, Republic of Korea

**Abstract** Neural crest progenitors are specified through the modulation of several signaling pathways, among which the activation of Wnt/β-catenin signaling by Wnt8 is especially critical. Glycoproteins of the Dickkopf (Dkk) family are important modulators of Wnt signaling acting primarily as Wnt antagonists. Here we report that Dkk2 is required for neural crest specification functioning as a positive regulator of Wnt/β-catenin signaling. Dkk2 depletion in *Xenopus* embryos causes a loss of neural crest progenitors, a phenotype that is rescued by expression of Lrp6 or β-catenin. Dkk2 overexpression expands the neural crest territory in a pattern reminiscent of Wnt8, Lrp6 and β-catenin gain-of-function phenotypes. Mechanistically, we show that Dkk2 mediates its neural crest-inducing activity through Lrp6 and β-catenin, however unlike Wnt8, in a GSK3β independent manner. These findings suggest that Wnt8 and Dkk2 converge on β-catenin using distinct transduction pathways both independently required to activate Wnt/β-catenin signaling and induce neural crest cells.

DOI: https://doi.org/10.7554/eLife.34404.001

*For correspondence:
jsj4@nyu.edu

Competing interests: The authors declare that no competing interests exist.

## Introduction

The neural crest is a population of cells unique to the vertebrate embryo. They are induced at the neural plate border during gastrulation, and around the time of neural tube closure, leave the neuro-epithelium to produce a diverse array of cell types, contributing to multiple tissues, including the heart, the peripheral nervous system, and the craniofacial skeleton.

Neural crest cells are generated through a sequence of events orchestrated by the modulation of several signaling pathways and the activation of a complex network of transcription factors (*Meulemans and Bronner-Fraser, 2004*; *Simões-Costa and Bronner, 2015*). A large body of evidence in several organisms indicates that attenuation of bone morphogenetic (BMP) signaling in conjunction with activation of Wnt/β-catenin signaling is critical to specify the neural crest (*Stuhlmiller and García-Castro, 2012*; *Bae and Saint-Jeannet, 2014*). Canonical Wnt ligands bind to Frizzled (Fzd) receptors and low-density-lipoprotein-related protein (Lrp5/6) co-receptors, which signals through the cytosolic adaptor protein Disheveled leading to inhibition of glycogen synthase kinase 3 (GSK3) and subsequent stabilization of β-catenin. β-catenin then translocates to the nucleus and in association with Tcf/Lef transcription factors activates Wnt target genes (*Clevers and Nusse, 2012*; *MacDonald et al., 2009*). Interfering with any components of Wnt/β-catenin signaling pathway blocks neural crest formation in the embryo.

The Dickkopf (Dkk) family of secreted glycoproteins acts primarily as negative modulators of Wnt signaling, this is especially true for Dkk1 and Dkk4 (*Niehrs, 2006*). Dkks interact with the Wnt co-receptors Lrp5/6, disrupting the binding of Lrp5/6 to the Wnt/Fzd ligand-receptor complex, thereby locally inhibiting Wnt regulated processes (*Niehrs, 2006*). Dkk1 was first identified for its ability to

inhibit Wnt signaling in *Xenopus* embryos and promote head formation (*Glinka et al., 1998*). Dkk1 injected embryos formed enlarged heads, and injection of Dkk1 blocking antibodies resulted in microcephalic *Xenopus* embryos (*Carmona-Fontaine et al., 2007*). Similarly, Dkk1 deficient mouse embryos lacked most head structures anterior of the otic vesicle (*Mukhopadhyay et al., 2001*). During neural crest formation, Dkk1 is expressed in the prechordal mesoderm and has been proposed as the inhibitory signal that prevents neural crest formation anteriorly by blocking Wnt/β-catenin signaling (*Carmona-Fontaine et al., 2007*). The role of Dkk2 is not as well defined. It has been proposed that Dkk2 can either activate or inhibit the pathway, depending on cellular context (*Wu et al., 2000*; *Li et al., 2002*; *Brott and Sokol, 2002*; *Li et al., 2005*; *Mukhopadhyay et al., 2006*), however its role during neural crest development has not been studied.

Here we show that Dkk2 knockdown prevents neural crest formation in vivo and in neuralized animal cap explants injected with Wnt8. Furthermore, Dkk2 gain-of-function increases the neural crest progenitor pool, reminiscent of Wnt8, Lrp6 and β-catenin gain-of-function phenotypes. We demonstrate that Dkk2 mediates its neural crest-inducing activity by activation Wnt/β-catenin signaling in a GSK3β independent manner. We propose that during neural crest formation, Lrp6 mediates two independent signaling events triggered by Wnt8 and Dkk2 converging on β-catenin to specify the neural crest.

## Results

### Dkk2 is required for neural crest formation

To evaluate Dkk2 function in the context of neural crest development we performed knockdown of Dkk2 using morpholino antisense oligonucleotides (MOs). A Dkk2MO was designed to specifically interfere with translation of *dkk2* mRNA (*Figure 1a*). The specificity of the MO was confirmed by Western blot of embryos injected with *Dkk2-Flag* mRNA and increasing doses of MO (*Figure 1b*). Unilateral injection of Dkk2MO in the animal region of 2 cell stage embryos resulted in a severe reduction of expression of two neural crest-specific genes *snai2* and *sox10* in a majority of injected embryos (*Figure 1c,d*). Concomitant with the loss of these genes the neural plate expression of *sox2* appeared broader on the injected side (*Figure 1c,d*). To confirm the specificity of Dkk2 knockdown phenotype, we used a second MO (Dkk2SMO) that specifically interferes with *dkk2* pre-mRNA splicing by targeting the intron 1-exon 2 junction (*Figure 1e*), resulting in the production of a longer transcript, due to intron 1 retention (*Figure 1f*; see also *Figure 1—figure supplement 1*). The phenotype of Dkk2SMO-injected embryos was identical to the phenotype generated by injection of the translation blocking MO, with *snai2* and *sox10* reduction and *sox2* expansion (*Figure 1g,h*). *snai2* expression in morphant embryos was efficiently rescued by injection of 10 pg of *Xenopus* Dkk2 plasmid DNA further establishing the specificity of the phenotype (*Figure 1—figure supplement 2*). Later in development, morphant embryos had a marked decrease in the number of melanoblasts visualized by the expression of *dct* (*Figure 1i,j*), and exhibited reduced craniofacial cartilages (*Figure 1k,l*) indicating that multiple neural crest-derivatives are affected in these embryos. The similarity of the two MO knockdown phenotypes, which are interfering with *dkk2* translation and splicing respectively, provides strong evidence for a specific requirement of Dkk2 during neural crest formation in vivo.

We expanded our analysis to include a broader repertoire of genes expressed at the neural plate border, including the neural plate border specifiers *pax3*, *snai1* and *sox8* as well as neural crest specifiers, *sox9* and *twist1* (*Meulemans and Bronner-Fraser, 2004*; *Hong and Saint-Jeannet, 2007*). We found that the neural border specifiers *pax3*, *snai1* and *sox8* were only mildly affected in Dkk2-depleted embryos, their expression level was largely unchanged however their expression domain appeared to be shifted laterally (*Figure 2a,c*). In contrast, the expression of the neural crest specifier *twist1* was downregulated, in a manner comparable to the phenotype observed for *sox10* and *snai2*, while *sox9* expression domain was either reduced or shifted laterally (*Figure 2a,c*). Furthermore the expression of the epidermal marker, *krt,* was reduced in a pattern consistent with the expansion of the neural plate (*Figure 2b,c*). Finally, the expression of the mesoderm markers *myod* and *actc1* was unchanged in Dkk2-depleted embryos ruling out possible secondary effects. The protocadherin, *pcdh8*, which is more broadly expressed in the mesoderm was also largely unaffected although its expression was shifted anteriorly in a subset of morphant embryos (*Figure 2c and d*).

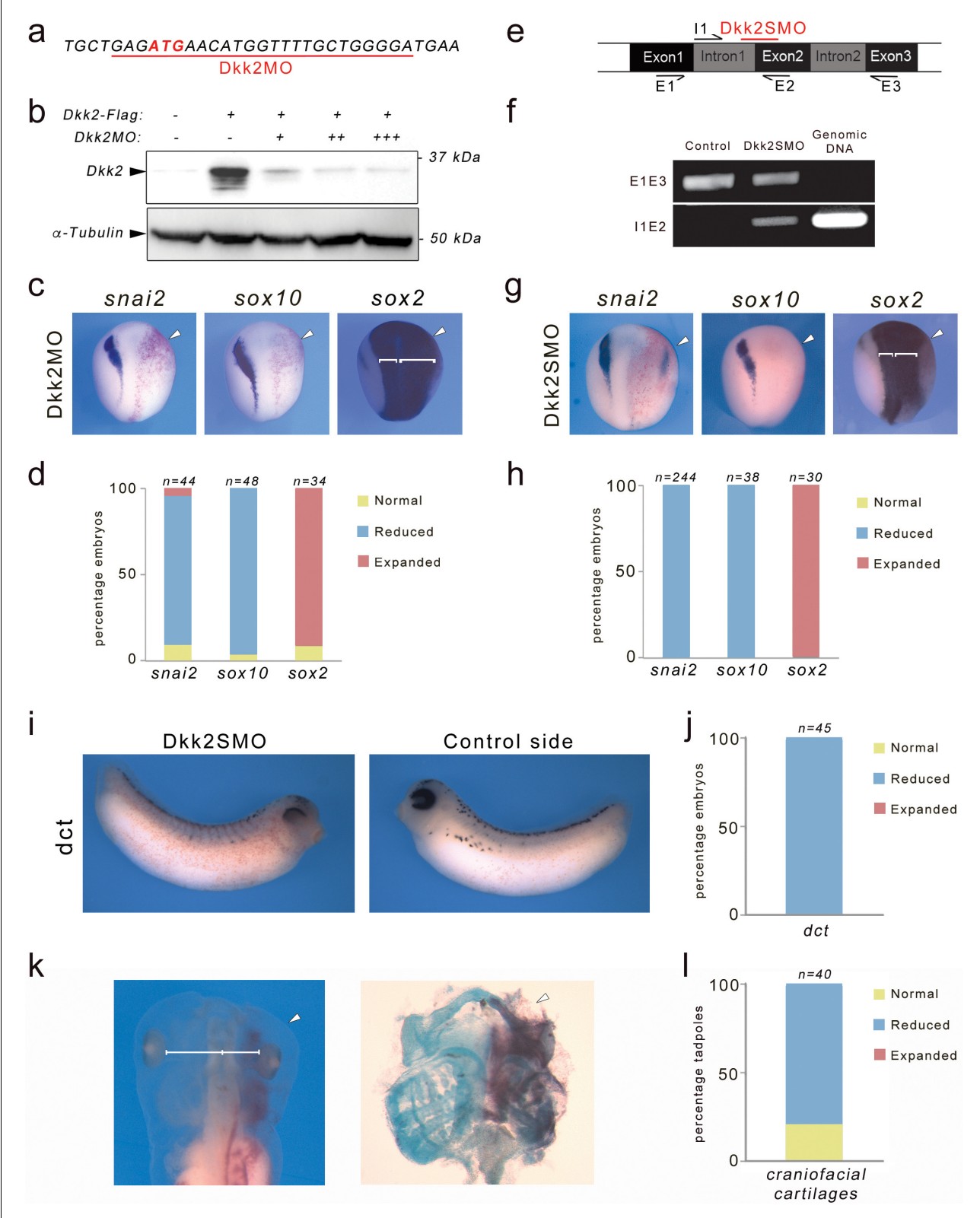

**Figure 1.** Dkk2 knockdown blocks neural crest formation in vivo. (**a**) The translation blocking MO (Dkk2MO) targets the initiation codon. (**b**) Western blot using lysates from embryos injected with *Dkk2-Flag* mRNA (10 ng) alone or in combination with increasing amounts of Dkk2MO, 2 ng (+), 5 ng (++), and 10 ng (+++), shows that Dkk2MO blocks Dkk2 protein accumulation in vivo. α-tubulin is shown as a loading control. (**c**) Unilateral injection of Dkk2MO (30 ng) at the 2 cell stage caused a reduction of *snai2* and *sox10* expression and a lateral expansion of *sox2* expression domain (brackets). The

*Figure 1 continued on next page*

*Figure 1 continued*

injected side (arrowheads) is to the right as indicated by the presence of the lineage tracer (Red-Gal). Dorsal views, anterior to top. (d) Quantification of the Dkk2MO phenotype. (e) Schematic representation of the dkk2 locus. The PCR primers used for the analysis of spliced transcripts are indicated. The position of the splice (Dkk2SMO) blocking MO is shown (red). (f) In Dkk2SMO-injected embryos a larger *dkk2* transcript is detected due to intron 1 retention. For all samples, the RT-PCR was performed under the same experimental conditions. (g) Unilateral injection of Dkk2SMO (30 ng) also resulted in a reduction of *snai2* and *sox10* expression and a lateral expansion of *sox2* expression domain (brackets). The injected side (arrowheads) is to the right as indicated by the presence of the lineage tracer (Red-Gal). Dorsal views, anterior to top. (h) Quantification of the Dkk2SMO phenotype. (i) At stage 30, Dkk2SMO-injected embryos show reduced *dct* expression. Lateral views, dorsal to top. (j) Quantification of the Dkk2SMO phenotype. (k) At stage 45, Dkk2SMO-injected tadpoles (20 ng) have reduced craniofacial structures (left panel). The white line indicates the distance between the brain and the eyes. These tadpoles have reduced craniofacial cartilages as revealed by alcian blue staining (right panel). In both panels the injected side (arrowheads) is to the right as indicated by the presence of the lineage tracer (Red-Gal). Ventral view, anterior to top. (l) Quantification of the results from three independent experiments. In all the graphs (d, h, j, l), the number of embryos analyzed (n) is indicated on the top of each bar.

DOI: https://doi.org/10.7554/eLife.34404.002

The following source data and figure supplements are available for figure 1:

**Source data 1.** Quantification of Dkk2 knockdown phenotype.

DOI: https://doi.org/10.7554/eLife.34404.005

**Figure supplement 1.** Quantification of unspliced vs. spliced transcripts in MO-injected embryos.

DOI: https://doi.org/10.7554/eLife.34404.003

**Figure supplement 2.** Dkk2 plasmid DNA injection rescues *snai2* expression in morphant embryos.

DOI: https://doi.org/10.7554/eLife.34404.004

Altogether these results suggest that Dkk2 does not participate in neural plate border specification but rather plays a role in neural crest progenitors formation and/or maintenance.

We also tested the function of Dkk2 in an animal cap explant assay. Activation of the Wnt/β-catenin pathway in conjunction with attenuation of BMP signaling induces neural crest genes in animal cap explants (*Figure 3a*; *Saint-Jeannet et al., 1997*; *LaBonne and Bronner-Fraser, 1998*). We found that the induction of *snai2* by Wnt8 and noggin (a BMP antagonist) was significantly repressed in Dkk2-depleted (Dkk2SMO-injected) animal cap explants, while co-injection of a control MO (CoMO) had no effect on *snai2* induction (*Figure 3b*). The reduction in *snai2* expression in these explants was associated with a significant increase in *sox2* expression, indicative of a loss of neural crest fate through inhibition of Wnt/β-catenin signaling (*Figure 3c*). Consistent with this activity, Dkk2 depletion completely blocked the induction of the Wnt-responsive TOP-FLASH reporter by expression of Wnt8 in animal cap explants (*Figure 3d*). Importantly, Dkk2 knockdown had no significant effect on the induction of neural tissue (*sox2*) by BMP inhibition or mesoderm (bra) by fibroblast growth factor 8b (FGF8b) in these explants (*Figure 3e,f*), further suggesting that Dkk2 acts specifically in Wnt/β-catenin signaling pathway. Altogether, these results indicate that Dkk2 is critical for Wnt-mediated neural crest induction in both the whole embryo and animal cap explants, thereby, positioning Dkk2 as a key regulator of neural crest specification in *Xenopus*.

## Developmental expression of Dkk2

We next analyzed the expression of *dkk2* during neural crest development. By qRT-PCR, *dkk2* is maternally expressed and *dkk2* transcripts start to accumulate at gastrulation (NF stage 12.5/13), consistent with a potential role in neural crest induction. This expression increases over time to reach a maximum around stage 20 and then progressively declines (*Figure 4a*). By in situ hybridization *dkk2* is broadly expressed at these stages, and does not appear to be distinctly enriched dorsally around the time of neural crest specification (*Figure 4b*). Later in embryogenesis (NF stage 40) *dkk2* accumulates in the gills and the developing heart (*Wu et al., 2000*). While *dkk1* is enriched anteriorly at the neurula stage, *dkk2* appears to be more abundant posteriorly (*Figure 4c*; *Carmona-Fontaine et al., 2007*). Altogether this expression suggests a broad requirement for Dkk2 during Wnt/β-catenin signaling. Therefore the regionalized expression of other components of the pathway, such as Fzd receptors and Wnt ligands, is likely to provide the spatiotemporal cues to achieve a localized response.

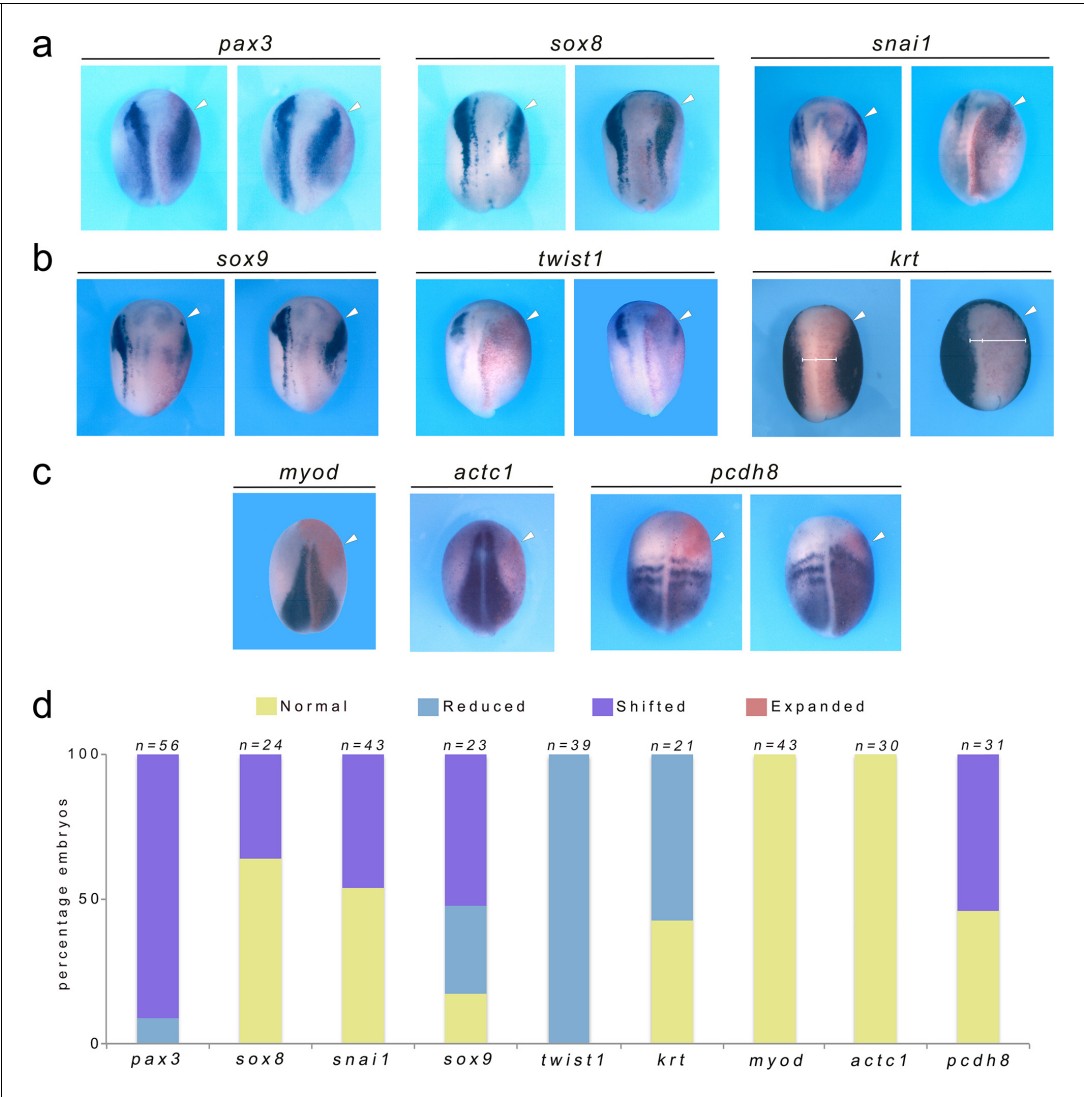

**Figure 2.** Dkk2 knockdown does not affect the expression of neural plate border specifiers and mesoderm formation. (a). Unilateral injection of Dkk2SMO (20 ng) did not affect the expression levels of *pax3*, *sox8* and *snai1*, although their expression was shifted laterally. (b) The neural crest specifier *twist1* was reduced, while *sox9* expression was shifted laterally in most embryos. The epidermal marker, *krt,* was reduced in a pattern consistent with the expansion of the neural plate. (c) The expression levels of the mesoderm markers *myod, actc1 and pcdh8* were unchanged in Dkk2-depleted embryos, although *pcdh8* expression domain was shifted anteriorly in a subset of morphant embryos. (a–c) Dorsal views, anterior to top. (d) Quantification of the Dkk2SMO phenotype. The number of embryos analyzed (n) is indicated on the top of each bar.

DOI: https://doi.org/10.7554/eLife.34404.006

The following source data is available for figure 2:

**Source data 1.** Quantification of Dkk2 knockdown phenotype.

DOI: https://doi.org/10.7554/eLife.34404.007

## Lrp6 and β-catenin can rescue neural crest formation in Dkk2-depleted embryos

In order to firmly establish that Dkk2 is functioning in Wnt/β-catenin signaling pathway during neural crest formation we analyzed the ability of Wnt8, Lrp6 or β-catenin to restore neural crest formation in Dkk2-depleted embryos. Activation of Wnt signaling before the mid-blastula transition (MBT) results in axis duplication, while Wnt activation post-MBT expands neural crest progenitors (*Saint-Jeannet et al., 1997*; *LaBonne and Bronner-Fraser, 1998*). For these experiments we injected plasmid DNA, which becomes transcribed only after MBT. We found that injection of β-catenin or Lrp6

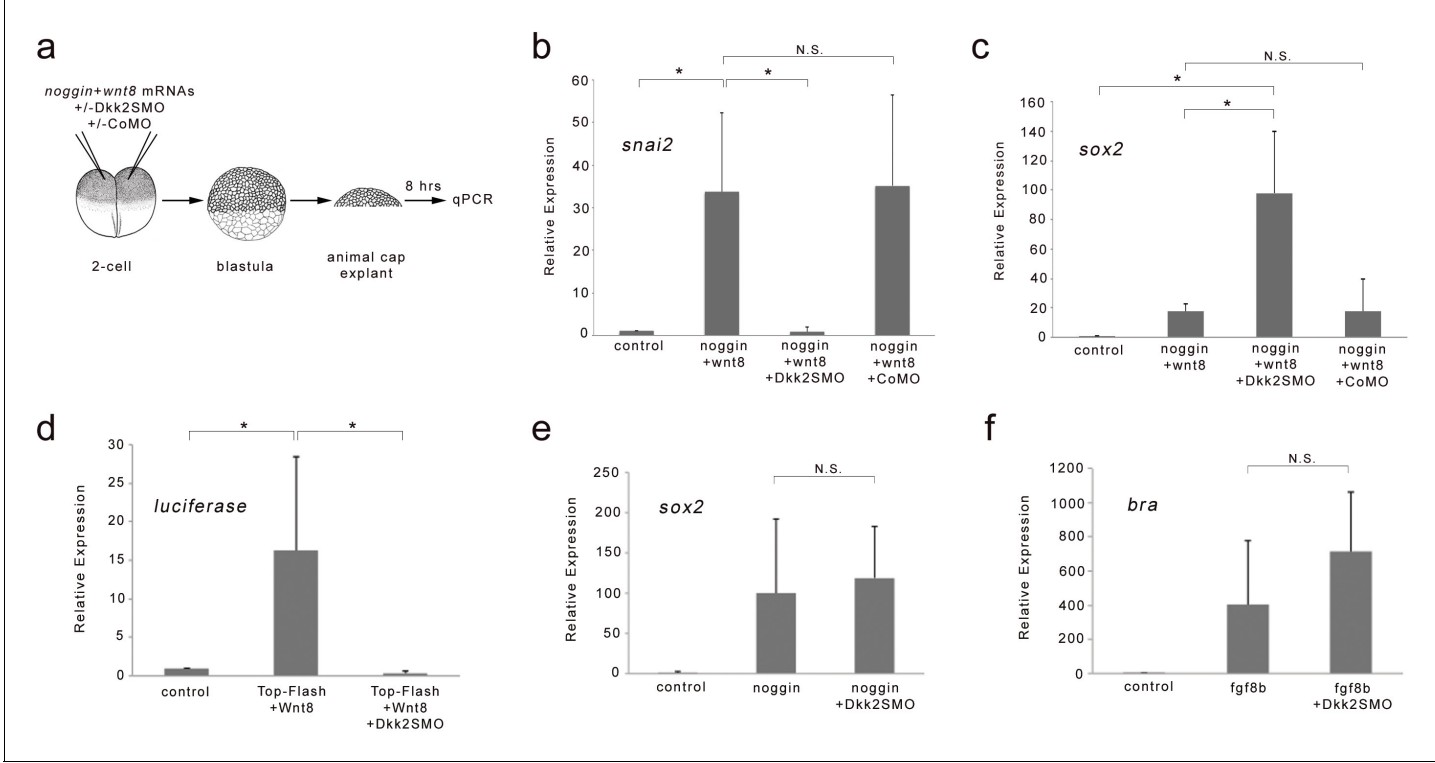

**Figure 3.** Dkk2 knockdown blocks neural crest induction by Wnt8 in neuralized animal cap explants. (a) At the 2 cell stage, mRNAs encoding *noggin* (50 pg) and *wnt8* (100 pg) were injected in the animal pole region alone or in combination with Dkk2SMO (30 ng) or CoMO (30 ng). At the blastula stage (NF stage 9), animal cap explants were dissected and cultured for 8 hr and analyzed by qRT-PCR. (b–c) Attenuation of Bmp signaling in combination with Wnt8 induces *snai2* expression. Dkk2SMO blocks *snai2* induction by Wnt8 to promote neural plate fate (*sox2* expression). A CoMO had no effect on the neural crest-inducing activity of Wnt8. (d) *wnt8* (100 pg mRNA) expression activates a TOP-FLASH reporter (10 pg DNA) construct in animal cap explants, an activity that is completely blocked by Dkk2SMO coinjection (30 ng). (e) The induction of the neural plate gene *sox2* by noggin (50 pg mRNA), and (f) the induction of the mesoderm gene *bra* by Fgf8b (100 pg mRNA) were unaffected by Dkk2SMO injection (30 ng). Graph represents mean ± s.e.m. of 3 independent experiments. *p<0.03; paired two tailed Student's t-test. n.s. not significant.
DOI: https://doi.org/10.7554/eLife.34404.008

DNA was quite potent at restoring *snai2* expression in Dkk2-depleted embryos, while injection of Wnt8 DNA was less efficient (*Figure 5a,b*). This rescue assay indicates that these factors are functioning in the same pathway, and that Dkk2 is acting upstream of Lrp6 and β-catenin during neural crest formation.

## Dkk2 promote neural crest formation in the embryo but cannot substitute for Wnt8 activity in neuralized animal cap explants

We next analyzed the gain-of-function phenotype of Dkk2, by injection of *dkk2* mRNA (500 pg) or plasmid DNA (25 pg) in one cell at the 2 cell stage. Dkk2 overexpression in both cases resulted in a lateral expansion of *snai2* expression domain in the vast majority of the embryos (*Figure 6a,b*). A similar lateral expansion was also observed for *sox8*, *sox9* and *sox10* (not shown). This phenotype is very reminiscent of Wnt8, Lrp6 and β-catenin gain-of-function phenotypes (*Tamai et al., 2000*; *Hong et al., 2008*) suggesting that these factors are functioning in the same pathway. To demonstrate that this activity is not a unique feature of *Xenopus* dkk2, we injected zebrafish and human Dkk2 plasmid DNA in the embryo and found that both were also capable of expanding *snai2* expression domain, although at a lower frequency (*Figure 6c,d*). This is in contrast to the well-described activity of Dkk1, which blocks *snai2* expression when misexpressed in the embryo (*Figure 6c,d*; *Carmona-Fontaine et al., 2007*). Dkk2 overexpression had no impact on the expression of three genes expressed in the mesoderm *myod*, *actc1* and *pcdh8* (*Figure 6e,f*). We also tested the ability of Dkk2 to induce *snai2* in animal cap explants neuralized by *noggin*. Interestingly, in this context Dkk2 was

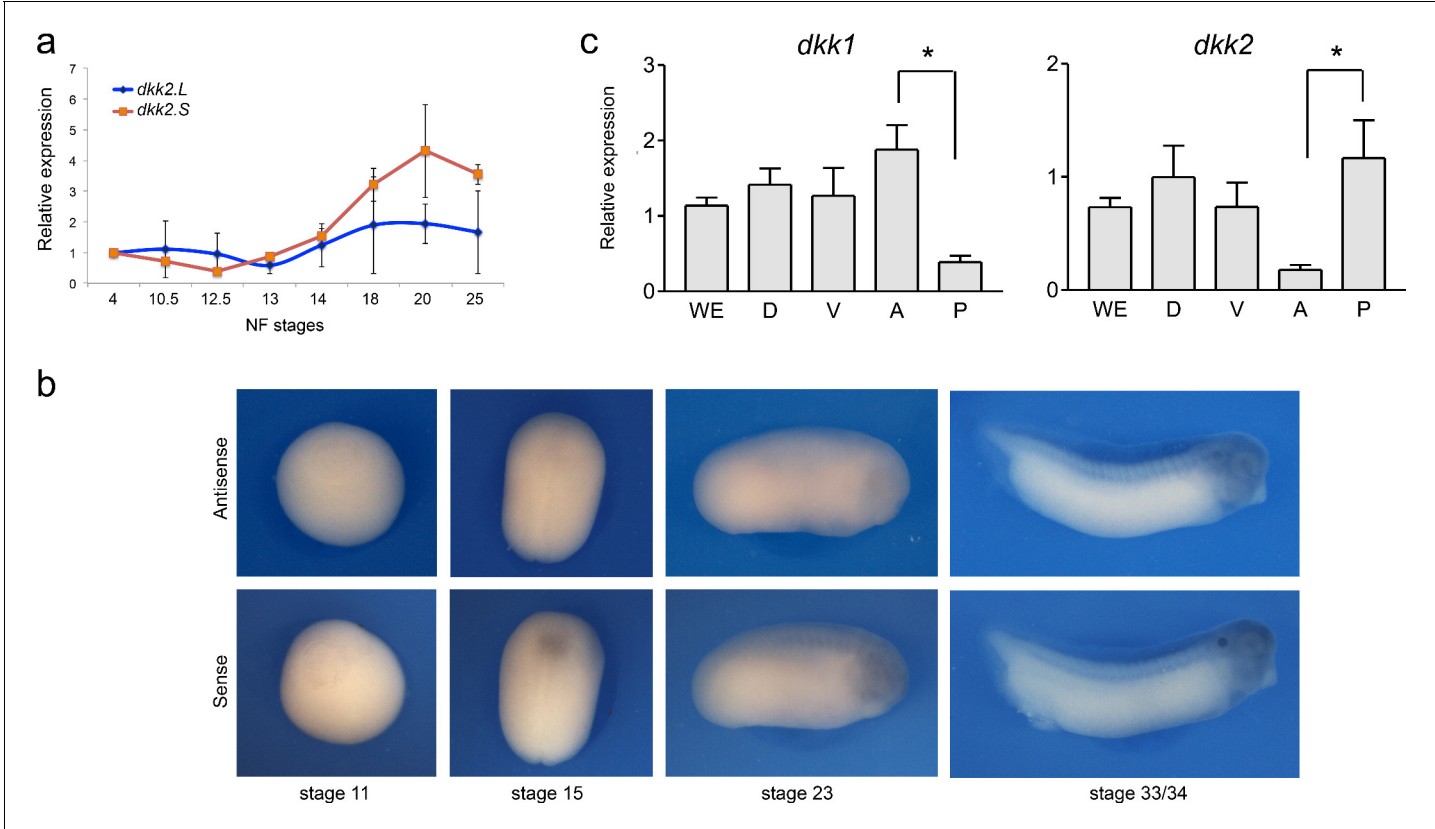

**Figure 4.** Developmental expression of *dkk2*. (**a**) Temporal expression of *dkk2.L* and *dkk2.S* by qRT-PCR. (**b**) By *in situ* hybridization, at all stages examined (NF stage 11-33/34) *dkk2* does not appear to be spatially restricted. Sense probe is shown as a control. (**c**) qRT-PCR analysis of *dkk1* and *dkk2* expression in dissected embryos at stage 15. WE; whole embryo; D, dorsal half; V, ventral half; A, anterior half; P, posterior half. The values were normalized to *Ef1a* and presented as mean ± s.e.m. * p<0.05, Student's t-test.

DOI: https://doi.org/10.7554/eLife.34404.009

unable to induce *snai2* (*Figure 6g*) indicating that Dkk2 is not functionally equivalent to Wnt8 in activating Wnt/β-catenin signaling pathway.

## Dkk2 neural crest-inducing activity requires active Wnt/β-catenin signaling

We next wished to determine whether Dkk2 mediated its neural crest-inducing activity in the embryo by activation of Wnt/β-catenin signaling. To test this we analyzed the ability of overexpressed Dkk2 to rescue *snai2* expression domain in the context of β-catenin-, Lrp6- or Wnt8-depleted embryos, using well-characterized MOs (*Heasman et al., 2000*; *Hassler et al., 2007*; *Park and Saint-Jeannet, 2008*). We found that in all three conditions Dkk2 was unable to restore *snai2* expression in these embryos (*Figure 7a,b*). These results indicate that β-catenin, Lrp6 and Wnt8 are all required for Dkk2 neural crest-inducing activity *in vivo*, and that Dkk2 induces neural crest via activation of Wnt/β-catenin signaling.

## Dkk2 activates Wnt/β-catenin signaling independently of GSK3β

Activation of Wnt/β-catenin signaling pathway is mediated through inhibition of GSK3β resulting in stabilization of β-catenin and its subsequent translocation to the nucleus to activate Wnt responsive genes. To gain further mechanistic insights into the activity of Dkk2 during neural crest induction, we compared the ability of BIO, a pharmacological GSK3-specific inhibitor (*Sato et al., 2004*), to rescue neural crest formation in Dkk2-depleted embryos and embryos overexpressing Dkk1. Control embryos treated with 10 µM BIO exhibited ectopic *sox10* expression anteriorly in a region where Wnt signaling is normally blocked by Dkk1 activity (*Figure 8a,b*; *Carmona-Fontaine et al., 2007*). In

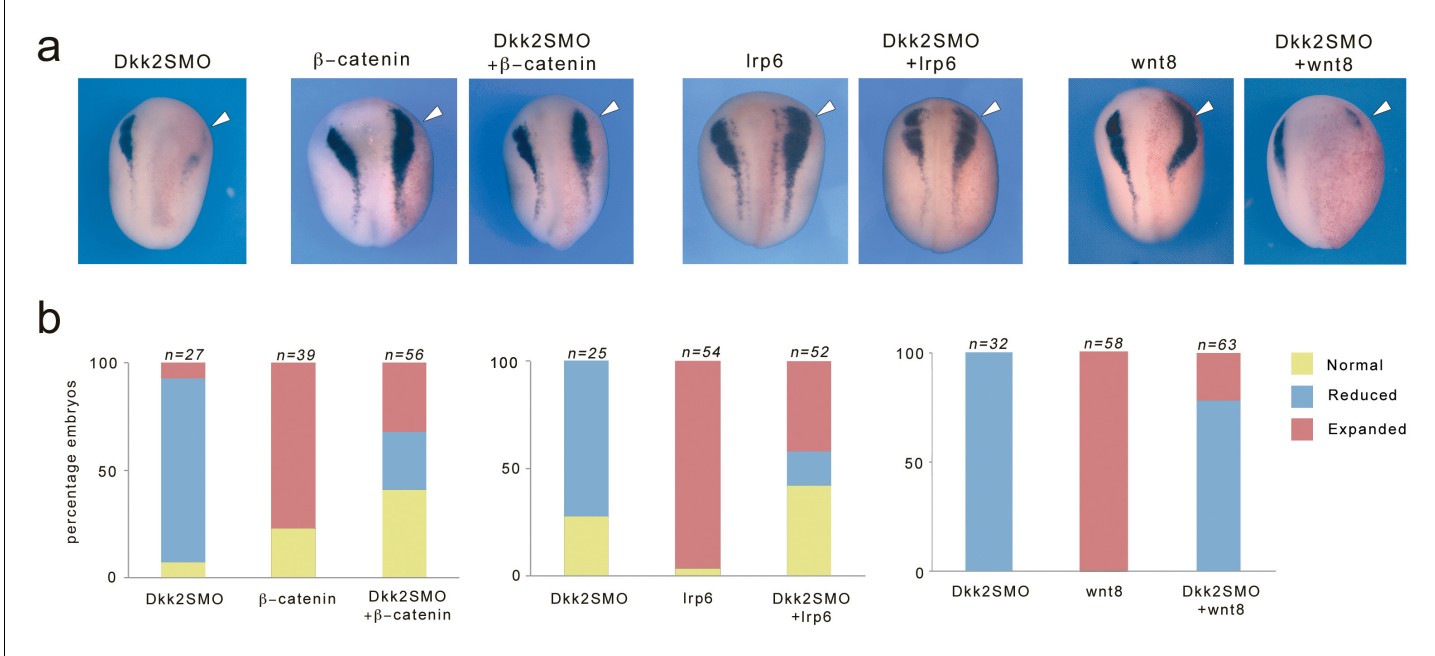

**Figure 5.** Expression of Lrp6 and β-catenin rescue neural crest formation in Dkk2-depleted embryos. (**a**) Unilateral injection of Dkk2SMO (30 ng) reduced *snai2* expression. This phenotype was efficiently rescued by injection of plasmid DNA encoding lrp6 (50 pg) or β-catenin (50 pg), and to a lesser extent by plasmid DNA encoding wnt8 (200 pg). Single injection of either plasmid DNA expanded *snai2* expression domain. The injected side (arrowheads) is to the right as indicated by the presence of the lineage tracer (Red-Gal). Dorsal views, anterior to top. (**b**) Quantification of the phenotypes. The number of embryos analyzed (n) is indicated on the top of each bar.

DOI: https://doi.org/10.7554/eLife.34404.010

The following source data is available for figure 5:

**Source data 1.** Quantification of Dkk2 knockdown phenotype upon β-catenin, lrp6 or wnt8 coexpression.

DOI: https://doi.org/10.7554/eLife.34404.011

Dkk1-injected embryos *sox10* expression was efficiently rescued, consistent with Dkk1 interference with Wnt-mediated GSK3β inhibition. By contrast BIO treatment was unable to restore *sox10* expression in the neural crest forming region of Dkk2-depleted embryos (*Figure 8a,b*). These observations further highlight the opposite function of these two Dkk family members, and indicate that Dkk2 mediates its neural crest-inducing activity through a branch of the Wnt/β-catenin signaling pathway that does not involve GSK3β inhibition.

## Discussion

It is widely accepted that Dickkopf proteins act as extracellular antagonists of Wnt/β-catenin signaling in development and cancer (*Niehrs, 2006*). In this study we show that one member of this family, Dkk2, acts in concert with Wnt ligands to activate Wnt/β-catenin signaling and promote neural crest formation. While overexpression studies have previously shown that Dkk2 can function as an activator of Wnt/β-catenin signaling when co-expressed with Fzd8 or Lrp6 (*Wu et al., 2000*; *Brott and Sokol, 2002*; *Li et al., 2002*), here we provide evidence for its positive role in a Wnt/β-catenin regulated developmental process. Dkk2 knockdown blocks and Dkk2 overexpression expands neural crest gene expression in the embryo. We show that Dkk2 mediates its neural crest-inducing activity through Lrp6 and β-catenin, however unlike Wnt8 in a GSK3β independent manner. We propose that during neural crest induction, Lrp6 mediates two independent signaling events triggered by Wnt8 and Dkk2, converging on β-catenin to promote neural crest formation (*Figure 9*).

One of the hallmarks of Dickkopf proteins is their ability to negatively modulate Wnt signaling, a well documented activity of Dkk1 (*Glinka et al., 1998*; *Semënov et al., 2001*; *Mukhopadhyay et al., 2001*). Similarly Dkk2 has been shown to regulate several developmental processes including eye, heart and palate development through its inhibitory function

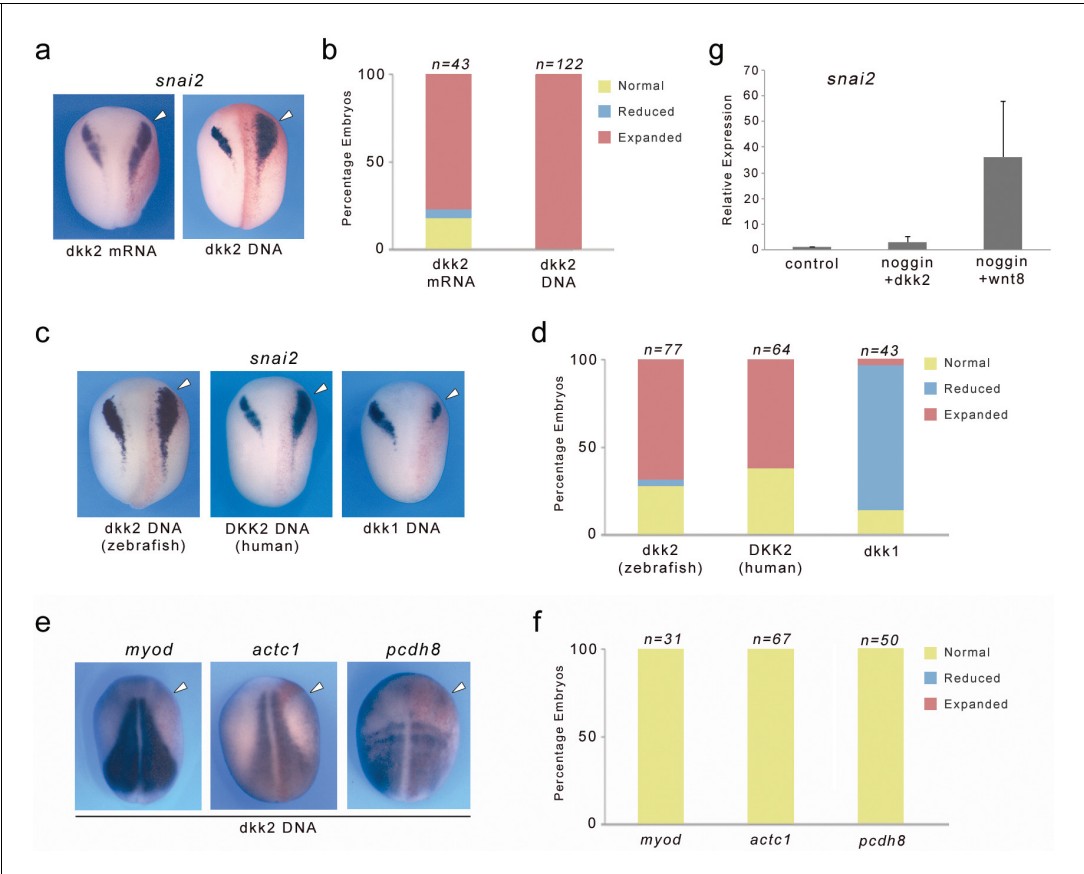

**Figure 6.** Dkk2 overexpression induces *snai2* expression in vivo but cannot substitute for Wnt8 activity in neuralized animal cap explants. (**a**) Unilateral injection of dkk2 mRNA (500 pg) or dkk2 plasmid DNA (25 pg) expanded *snai2* expression domain laterally. (**b**) Quantification of the Dkk2 overexpression phenotype. The number of embryos analyzed (n) is indicated on the top of each bar. (**c**) Zebrafish or Human *Dkk2* plasmid DNA injections also expanded *snai2* expression, while dkk1 overexpression blocked *snai2* expression. (**e**) The expression of the mesoderm markers *myod*, *actc1* and *pcdh8* was unchanged upon Dkk2 overexpression. (**d, f**) Quantification of the phenotypes. The number of embryos analyzed (n) is indicated on the top of each bar. (**a, c, e**) The injected side (arrowheads) is to the right as indicated by the presence of the lineage tracer (Red-Gal). Dorsal views, anterior to top. (**g**) Unlike *wnt8*, injection of *dkk2* mRNA (500 pg) is unable to induce *snai2* in animal cap explants neuralized by noggin.

DOI: https://doi.org/10.7554/eLife.34404.012

The following source data is available for figure 6:

**Source data 1.** Quantification of Dkk2 gain-of-function phenotype.

DOI: https://doi.org/10.7554/eLife.34404.013

(*Mukhopadhyay et al., 2006*; *Gage et al., 2008*; *Phillips et al., 2011*; *Li et al., 2017*). However, the situation for Dkk2 is somewhat more complex as its activity appears to be context dependent. For example, unlike Dkk1, Dkk2 does not promote the formation of enlarged heads when overexpressed in *Xenopus* embryo, it has in fact the opposite effect generating microcephalic and cyclopic embryos, similar to Wnt8 gain-of-function phenotype (*Wu et al., 2000*). Dkk2 is also a poor inhibitor of Wnt-8-induced axis duplication as compared to Dkk1 (*Krupnik et al., 1999*; *Wu et al., 2000*). Furthermore when overexpressed with Fzd8 or Lrp5/6, Dkk2 can activate Wnt/β-catenin signaling (*Wu et al., 2000*; *Brott and Sokol, 2002*; *Li et al., 2002*), and in *Xenopus* this activity can be blocked by Dkk1 expression (*Mao and Niehrs, 2003*). The molecular mechanism underlying these functional differences is not well understood. The Dkks are characterized by the presence of two conserved cysteine-rich domains, an N-terminal cysteine-rich domain, known as Dkk_N, and a C-terminal cysteine-rich domain, resembling a colipase fold (*Niehrs, 2006*). Structure function analyses using chimeric constructs and the axis duplication assay as readout have revealed that the different activities of Dkk1 and Dkk2 rest on the unique properties of their respective Dkk_N domains, while the C-terminal cysteine-rich domain primarily promotes Lrp6 signaling (*Brott and Sokol, 2002*).

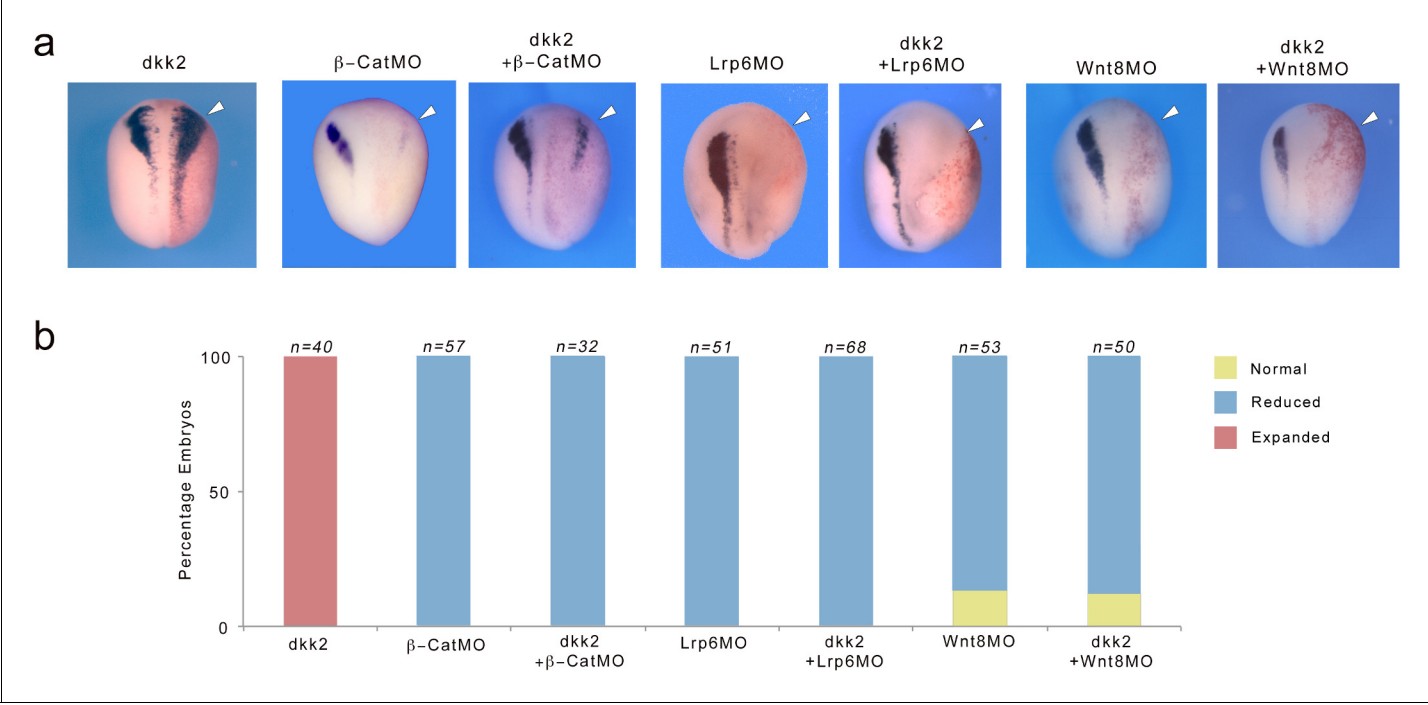

**Figure 7.** Dkk2 neural crest-inducing activity requires active Wnt/β-catenin signaling. (a) Unilateral injection of *dkk2* plasmid DNA (50 pg) expanded *snai2* expression domain laterally (arrowhead). This activity was blocked in the context of β-catenin- (β-catMO; 20 ng), Lrp6- (Lrp6MO; 20 ng) or Wnt8-(Wnt8MO; 40 ng) depleted embryos. The injected side (arrowheads) is to the right as indicated by the presence of the lineage tracer (Red-Gal). Dorsal views, anterior to top. (b) Quantification of the phenotypes. The number of embryos analyzed (n) is indicated on the top of each bar.
DOI: https://doi.org/10.7554/eLife.34404.014

The following source data is available for figure 7:

**Source data 1.** Quantification of β-catenin, Lrp6 and Wnt8 knockdown phenotypes upon Dkk2 coexpression.
DOI: https://doi.org/10.7554/eLife.34404.015

Similarly in cultured cells, the C-terminal region of Dkk2 inhibited Wnt3a signaling, but activated a Wnt-responsive promoter upon LRP6 co-expression (*Li et al., 2002*). Another study, has proposed a different mechanism by which the inhibitory activity of Dkk2 was primarily dependent on the presence of the Dkk receptor, kremen2, possibly by promoting internalization of the receptor complex, thereby preventing Lrp6 signaling. When kremen2 is absent, Dkk2 acts as an activator of the pathway (*Mao and Niehrs, 2003*).

Gain-of-function experiments indicate that Dkk2 is a potent inducer of neural crest genes in the embryos. Dkk2 requires Lrp6 and β-catenin to promote neural crest formation, suggesting that Dkk2 mediates its neural crest-inducing activity through activation of Wnt/β-catenin signaling. This activity is not a unique feature of *Xenopus* Dkk2, as Human and zebrafish Dkk2 were also capable of inducing neural crest genes in the embryo. Interestingly, Dkk2 was unable to induce neural crest formation in neuralized animal cap explants, indicating that Dkk2 in this context cannot substitute for Wnt8. This result suggests that signaling by Wnt8 and Dkk2 are likely to be concurrently required to activate the Wnt/β-catenin pathway during neural crest specification.

Dkk2-/- mutant mice are characterized by low bone density and osteopenia (*Li et al., 2005*). These animals have also a complete conversion of the corneal epithelium into stratified epithelium with epidermal characteristics. Analysis of a Wnt reporter (TOP-GAL) indicates that these mutants have increased Wnt/β-catenin signaling in the cornea (*Mukhopadhyay et al., 2006*). Our loss-of-function experiments point to a role of Dkk2 as a positive regulator of Wnt/β-catenin signaling. Using two distinct MOs interfering with Dkk2 mRNA translation and splicing we found that Dkk2 was required for neural crest induction in vivo and in animal cap explants. In these explants activation of the Wnt responsive TOP-FLASH reporter was also dependent on Dkk2 activity. The ability of Lrp6 and β-catenin to rescue Dkk2 knockdown phenotype, together with the fact that the GSK3 inhibitor

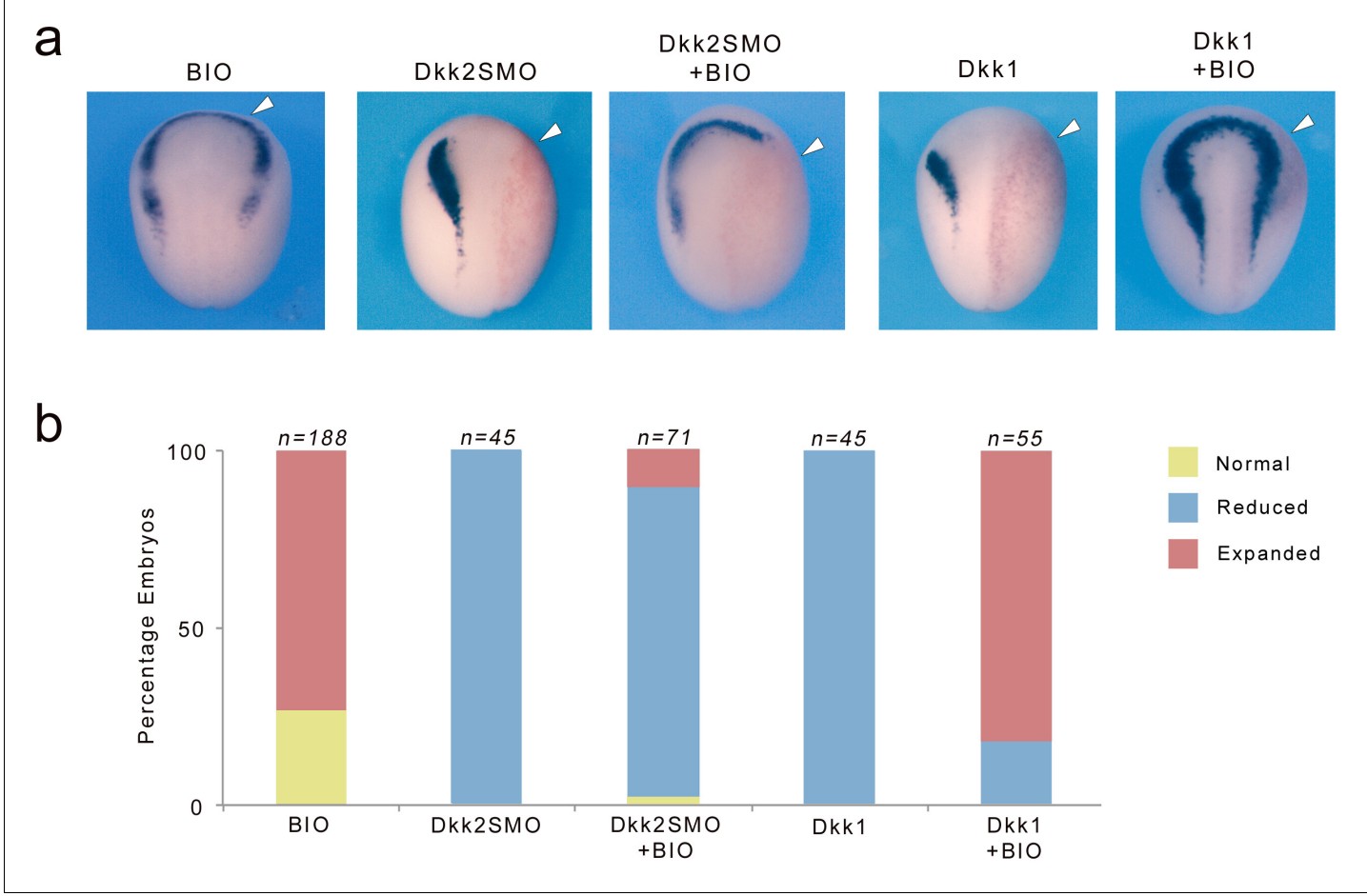

**Figure 8.** Dkk2 mediates its activity independently of GSK3. (a) BIO treatment (10 μM) expanded *sox10* expression domain anteriorly. Unilateral injection of Dkk2SMO (30 ng) reduced *sox10* expression a phenotype that cannot be rescued by BIO treatment. In contrast the Wnt inhibitory activity of Dkk1 (50 pg) on *sox10* expression was efficiently rescued by BIO treatment. The injected side (arrowheads) is to the right as indicated by the presence of the lineage tracer (Red-Gal). Dorsal views, anterior to top. (b) Quantification of the phenotypes. The number of embryos analyzed (n) is indicated on the top of each bar.

DOI: https://doi.org/10.7554/eLife.34404.016

The following source data is available for figure 8:

**Source data 1.** Quantification of Dkk2 knockdown and Dkk1 overexpression phenotypes upon BIO treatment.
DOI: https://doi.org/10.7554/eLife.34404.017

(BIO) failed to restore neural crest gene expression in these embryos, strongly suggest that Dkk2 mediates its activity through Lrp6 and β-catenin, and point to a GSK3β independent function of Dkk2 during neural crest formation. These results are consistent with another study suggesting a DVL and GSK3β independent pathway in the activation of Wnt signaling by Dkk2 and Lrp6 in trans-fected cells (*Li et al., 2002*).

It is likely that this activity of Dkk2 as a positive regulator of Wnt/β-catenin signaling will apply to other biological contexts. For example Dkk1 and Dkk2 have been shown to have opposite functions in regulating angiogenesis, however it is still unclear whether Dkk2 mediates its activity through the Wnt pathway in this context. Interestingly, we were able to extend our observations (*Figure 9*) to another Wnt regulated process, axis duplication by Wnt8; (*Sokol et al., 1991*; *Smith and Harland, 1991*), demonstrating that Wnt8's ability to induce secondary axis was directly dependent on Dkk2 function (*Figure 9—figure supplement 1*). Further studies are needed to identify the downstream effectors of the alternate pathway activated by the Lrp6/Dkk2 complex to understand the precise

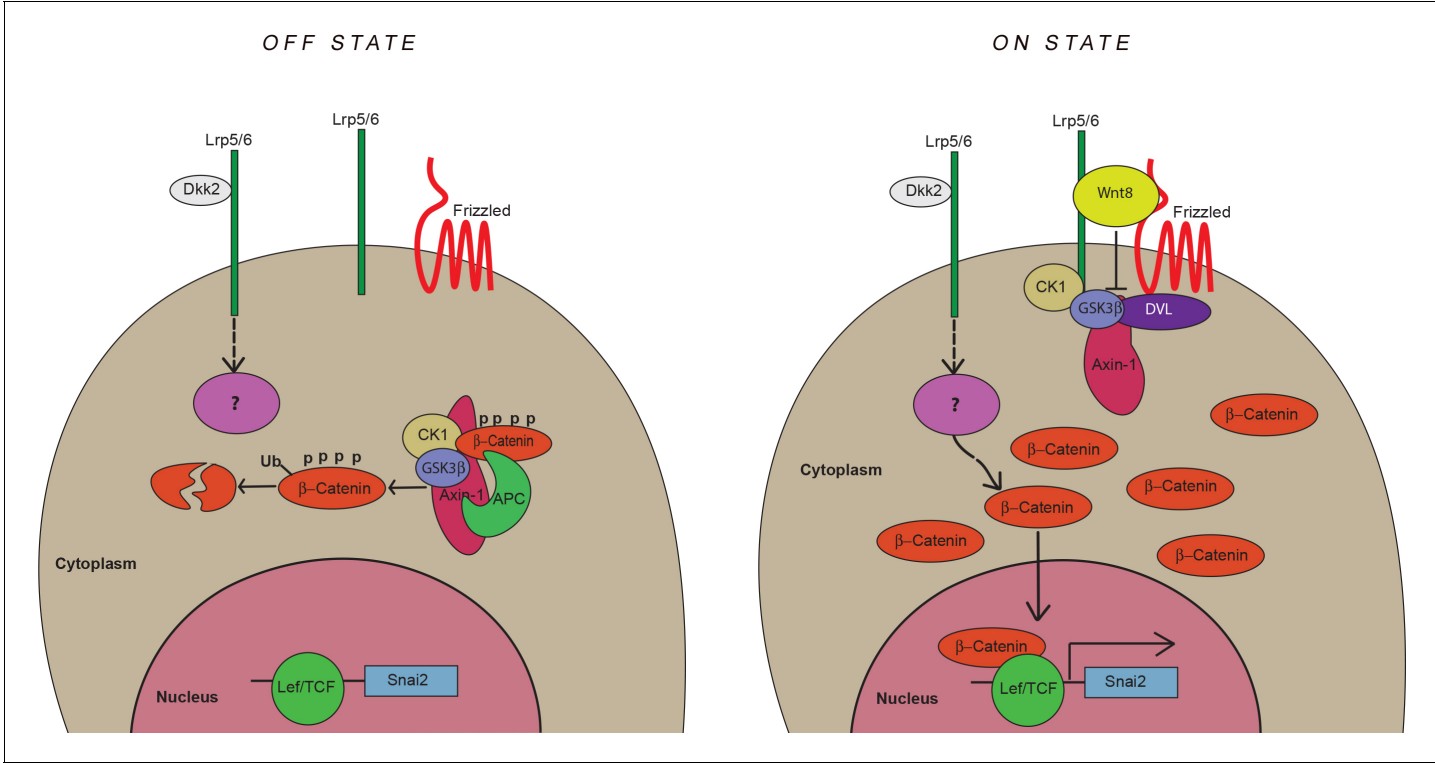

**Figure 9.** Model for neural crest induction by Dkk2 and Wnt8. During neural crest induction, Lrp6 mediates two independent signaling events triggered by Wnt8 and Dkk2 converging on β-catenin to promote neural crest formation (*snai2* induction). The Lrp6/Wnt/Fzd complex signals through disheveled (DVL) leading to inhibition of GSK-3β and stabilization of β-catenin. The Lrp6/Dkk2 complex signals through an alternate pathway converging on β-catenin. The components of this alternate pathway are unknown.

DOI: https://doi.org/10.7554/eLife.34404.018

The following figure supplement is available for figure 9:

**Figure supplement 1.** Dkk2 function is required for dorsal axis duplication by Wnt8.

DOI: https://doi.org/10.7554/eLife.34404.019

mechanism of Dkk2-mediated activation of Wnt signaling, and define the underlying mechanisms that contribute to its context dependent activity.

# Materials and methods

## Plasmids and morpholinos

Expression constructs for *Xenopus*, zebrafish and human Dkk2 were generous gifts of Dr. Christof Niehrs (University of Mainz, Mainz, Germany), Dr. Tatiana Piotrowski (Stowers Institute, MO, USA) and Dr. Sergei Sokol (Mount Sinai, NY, USA), respectively. A flagged version of Xenopus Dkk2 was generated by adding a Flag tag (DYKDDDDK) at the C-terminal using the following primers F: GAA TTCGCCACCGAGATGAACATGGTTTTGCTGGGGA; R: ctcgagTTACTTGTCGTCGTCGTCCTTGTAG TCTATTTTCTGGCAAATG. The PCR product was cloned into pCS2+ (pCS2+ Dkk2 Flag). Control (CoMO), Dkk2 (Dkk2MO: TCCCCAGCAAAACCATGTTCATCTC; Dkk2SMO: GGAATGCAAATGCC TACAAGATATA), Lrp6 (Lrp6MO; *Hassler et al., 2007*), β-Catenin (β-CatMO; *Heasman et al., 2000*), Wnt8 (Wnt8MO; *Park and Saint-Jeannet, 2008*) morpholino antisense oligonucleotides (MO), were purchased from GeneTools (Philomath, OR). The specificity of the translation blocking MO (Dkk2MO) was tested on Western blot of embryos injected with a Dkk2-Flag mRNA and increasing doses of MO (*Figure 1a,b*). The splice blocking MO, Dkk2SMO, was validated by RT-PCR on injected embryos (*Figure 1e,f*) using the following primers E1: TAAGGAGTGTGAAGTTGGAAGG, E3: TTTGAAGAGTAGGTGGCATCTT; I1: AATATCTTCTTAGGGCCCAACTG and E2: GGTCTCAAG TGCTGGGATATG. The efficiency of Dkk2SMO was further established by qRT-PCR (*Figure 1—*

*figure supplement 1*) using the following primers E1: CACGGAGTCTCACACAAGAAA; E2: GACTG TAGCAGTACCTTCCAA; I2: CAGCACTCTACAGCAGAACAA; and I3: TCCTTCCTCTTGGCTC TTTAAC.

## Embryos, injections, and animal cap explants culture

*Xenopus laevis* embryos were staged according to *Nieuwkoop and Faber, 1967* and raised in 0.1X NAM (Normal Amphibian Medium; *Slack and Forman, 1980*). This study was performed in accordance with the recommendations of the Guide for the Care and Use of Laboratory Animals of the National Institutes of Health. The procedures were approved by New York University Institutional Animal Care and Use Committee, under animal protocol # 150201. *Xenopus wnt8* (25 pg; *Christian et al., 1991*), *noggin* (10 pg; *Smith and Harland, 1992*), *dkk2* (500 pg; *Wu et al., 2000*), *fgf8b* (25 pg; *Fletcher et al., 2006*) mRNAs were synthesized in vitro using the Ambion Message Machine kit (Austin, TX). For plasmid DNA, 25 pg (*dkk2*), 200 pg (*wnt8*) or 50 pg (*dkk1, lrp6* and *β-catenin*) was injected per embryo. MOs, mRNAs and plasmid DNA were injected in one blastomere at the 2 cell stage (NF stage 2) and embryos were analyzed by in situ hybridization at the neurula stage (NF stage 14–17). To identify the injected side, 500 pg of β-galactosidase mRNA was coinjected as a lineage tracer. Only embryos with co-localized expression of the lineage tracer with the cell type marker were considered for analysis. Control and injected embryos were treated in the dark with 10 µM of GSK3 inhibitor (BIO; Sigma-Aldrich, St Louis MO) at the gastrula stage (NF stage 11/11.5), and collected at NF stage 14–17. For the axis duplication assay, embryos were injected in the equatorial region in both ventral blastomeres at 4 cell stage (NF stage 3) and analyzed at NF stage 32. For animal cap explant experiments, both blastomeres at the 2 cell stage (NF stage 2) were injected in the animal pole region. Explants were dissected at the blastula stage (NF stage 9) and cultured for 8 hr in NAM 0.5X. In rescue experiments, the injections of mRNAs/plasmid DNA and MOs were performed sequentially. For whole embryo injections and animal cap explant assays each experiment was performed on at least three independent batches of embryos.

## Lineage tracing, whole-mount in situ hybridization and cartilage staining

Embryos at the appropriate stage were fixed in MEMFA and stained for Red-Gal (Research Organics; Cleveland, OH) to visualize the lineage tracer (β-gal mRNA) on the injected side and processed for in situ hybridization. Antisense digoxygenin-labeled probes (Genius kit; Roche, Indianapolis IN) were synthesized using template cDNA encoding snai1 (*Essex et al., 1993*), snai2 (*Mayor et al., 1995*), sox8 (*O'Donnell et al., 2006*), sox9 (*Spokony et al., 2002*), sox10 (*Aoki et al., 2003*), sox2 (*Mizuseki et al., 1998*), pax3 (*Bang et al., 1997*), twist1 (*Hopwood et al., 1989a*), krt (*Jonas et al., 1985*), myod (*Hopwood et al., 1989b*), actc1 (*Mohun et al., 1984*), pcdh8 (*Kim et al., 1998*) and dct (*Aoki et al., 2003*). Whole-mount in situ hybridization was performed as described (*Harland, 1991*; *Saint-Jeannet, 2017*). Cartilage staining was performed on stage 45 tadpole heads as previously described (*Devotta et al., 2016*).

## Western blot analysis

Embryos were injected at the 4 cell stage with 10 ng of *Xenopus* Dkk2-Flag mRNA alone or in the presence of increasing doses of Dkk2MO, and cultured to stage 13. Pools of 10 embryos were homogenized in lysis buffer (0.5% Triton X-100, 10 mM Tris–HCl at pH 7.5, 50 mM NaCl, 1 mM EDTA), containing Halt[TM] Protease Inhibitor Cocktail (ThermoFisher Scientific; Waltham, MA). After two consecutive centrifugations to eliminate lipids, the lysate was concentrated on an Amicon Ultra Centrifugal Filter (Merck Millipore; Billerica, MA), 5 µl of the concentrated lysate was resolved on a 10% NuPAGE Bis-Tris gel and transferred onto a PVDF membrane using the iBlot system (Invitrogen). Blots were subsequently incubated overnight with one of the following primary antibodies: monoclonal anti-Flag M2 antibody (Sigma Aldrich, F3165; 1:1000 dilution) and anti α-tubulin antibody (Sigma Aldrich, T9026; 1:500 dilution). The blots were then washed and incubated with anti-mouse IgG coupled to horseradish peroxidase (Santa Cruz Biotechnology; 1:10,000 dilution). Peroxidase activity was detected with the Western Blotting Luminol Reagent (Santa Cruz Biotechnology) and imaged on a ChemiDoc MP Biorad gel documentation system (Hercules, CA). Membranes were

stripped using Restore Western Blot Stripping Buffer (ThermoFisher Scientific) according to the manufacturer recommendations.

## qRT-PCR analysis

Total RNAs were extracted from embryos or animal cap explants using the RNeasy micro RNA isolation kit (Qiagen, Valencia, CA). The RNA samples were digested with RNase-free DNase I before RT-PCR. The amount of RNA isolated was quantified by measuring the optical density using a Nanodrop spectrophotometer (Nanodrop Technologies, Wilmington, DE). Approximately 250 ng of total RNAs from animal caps was reverse transcribed using the Superscript VILO cDNA Synthesis Kit (Invitrogen, Grand Island, NY) and 2 µl of 1:1000 dilution of the synthesized cDNA was amplified using *Power* SYBR Green PCR Master Mix (Applied Biosystems, Foster City, CA) on a QuantStudio 3 Real-Time PCR System (Applied Biosystems, Foster City, CA) with the following primer sets: *bra* (F: GAATGG TGGAGGCCAGATTAT; R: TCAGGGAATGAATGGCTAGTG), *sox2* (F: GCGTCCAACAACCAGAA TAAG; R: GTTCTCCTGAGCCATCTTTCT), *snai2* (F: AGGCACGTGAAGGGTAGAGA; R: CATGGGAA TAAGTGCAACCA), *luciferase* (F: GTGTTGGGCTTATTTATC; R: TAGGCTGCGAAATGTTCATACT) and *odc* (F: ACATGGCATTCTCCCTGAAG; R: TGGTCCCAAGGCTAAAGTTG). The PCR conditions were as follows: denaturation 95℃ (15 s), annealing and extension at 60℃ (1 min) for 40 cycles.

## Acknowledgements

We are grateful to Dr. Christof Nierhs (University of Mainz, Germany), Dr. Tatiana Piotrowski (Stowers Institute, USA) and Dr. Sergei Sokol (Mount Sinai, USA) for reagents. This work was supported by grants from the National Institutes of Health to J-P S-J (R01-DE014212 and R01-DE025468).

## Additional information

### Funding

| Funder | Grant reference number | Author |
|---|---|---|
| National Institutes of Health | DE025468 | Jean-Pierre Saint-Jeannet |
| National Institutes of Health | DE014212 | Jean-Pierre Saint-Jeannet |

The funders had no role in study design, data collection and interpretation, or the decision to submit the work for publication.

### Author contributions

Arun Devotta, Conceptualization, Formal analysis, Investigation, Methodology, Writing—original draft, Writing—review and editing; Chang-Soo Hong, Conceptualization, Formal analysis, Investigation, Methodology; Jean-Pierre Saint-Jeannet, Formal analysis, Investigation, Supervision, Funding acquisition, Methodology, Writing—review and editing

### Author ORCIDs

Jean-Pierre Saint-Jeannet https://orcid.org/0000-0003-3259-2103

### Ethics

Animal experimentation: This study was performed in accordance with the recommendations of the Guide for the Care and Use of Laboratory Animals of the National Institutes of Health. The procedures were approved by New York University Institutional Animal Care and Use Committee (IACUC), under animal protocol # 150201.

### Decision letter and Author response

Decision letter https://doi.org/10.7554/eLife.34404.024
Author response https://doi.org/10.7554/eLife.34404.025

## Additional files

### Supplementary files

• Transparent reporting form
DOI: https://doi.org/10.7554/eLife.34404.020

### Data availability

All data generated or analysed during this study are included in the manuscript and supporting files.

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
