## [Decision Letter]

Thank you for submitting your article "Dkk2 promotes neural crest specification by activating Wnt/β-catenin signaling in a GSK3β independent manner" for consideration by *eLife*. Your article has been reviewed by three peer reviewers, and the evaluation has been overseen by Marianne Bronner as the Senior and Reviewing Editor.

The reviewers have discussed the reviews with one another and the Reviewing Editor has drafted this decision to help you prepare a revised submission.

Summary:

This is an interesting paper that addresses the role of Dkk2 in neural crest formation, showing that it is required for the expression of both Snai2 and Sox10. Loss of Dkk2 can be rescued by β-catenin and Lrp6, as well as Wnt8. Over-expression of Dkk2 leads to an expansion of Snai2. While a GSK3β inhibitor expands Snai2 expression anteriorly, the prospective neural crest territory remains sensitive to Dkk2 MO, suggesting it functions in a GSK3β independent manner.

Essential revisions:

1) It is critical to provide better expression profile of Dkk2 and it is suggested to move the images to main figures. The published expression pattern for Dkk2 does not show expression in the neural fold or neural crest. Furthermore the RNA seq data available on Xenbase shows a very low expression of Dkk2 during neural crest induction. It is even more striking when the expression data is compared to the Wnt partner namely Wnt8 and Lrp6. It is only in the Discussion that the authors refer to expression data presented in the supplemental material.

2) The expression data suggest that Dkk2 is an extremely potent activator of the pathway that despite its minute expression it is essential for proper Wnt signaling. This is not entirely consistent with the fact that loss of Dkk2 can be rescued by overexpression of Wnt8, Lrp6 or β-catenin since all of these mRNA are expressed at 10 to 100 folds excess when compared to Dkk2. Does the Dkk2 protein possess some particular property such as extreme stability or extreme binding affinity to the co-receptors? Following a tagged Dkk2 protein in the embryo (by blot and IF) could provide some clues. Dose response in the embryo or animal cap could be used to determine the effectiveness of Dkk2 compare to Dkk1. What is the minimal dose at which Dkk2 expression can expend the neural crest?

3) While the results strongly suggest that Dkk2 potentiates WNT signaling towards neural crest formation, the mechanism of action is unresolved. The authors should assess whether Dkk2 triggers expression of known direct targets of WNT/β-cat pathway, or non-canonical WNT pathways. They should also test mesoderm markers to rule out possible secondary effects.

4) The authors should test a broader repertoire of neural crest markers (in addition to Sox10 and Snai2). In addition, analysis of early stages (st. 13) is required to test effects on early NC specification markers.

5) In the absence of lineage tracing, the loss of *snai2* and *sox10* expression is not conclusive evidence for loss of neural crest cells. The authors should examine non-neural ectoderm markers. They should also let embryos grow to later stages, as loss of neural crest should result in complete unilateral absence of craniofacial cartilage and cranial peripheral nervous system. If so, is this rescued by β-catenin and Lrp6 injections in Dkk2-depleted embryos?

6) While the loss of function phenotype is compelling, the in vitro translation does not allow one to test the effective amount to inject in an embryo. Blocking the translation of a tagged Dkk2 mRNA would help define that dose. In addition, rescuing Dkk2 MO with an insensitive mRNA would confirm the specificity of the phenotype. The splice morpholino need to be more carefully quantified. Using realtime qPCR what is the percentage of properly spliced mRNA with various doses of the MO?

Other suggested revisions:

1) The authors should references previous studies that provide molecular insights into how Dkk2 activates Wnt/β-cat. For example, Li et al. (2002) suggested that it is the C2 domain of Dkk2 that is critical in its WNT activating functions, and further provide evidence suggesting that Dkk2 WNT inductive role operates through DVL and GSK3 independent mechanisms. Mao and Niehrs (2003) suggested that Krm2 is responsible for the WNT-antagonistic effect of Dkk2, and that in its absence, Dkk2 operates as a WNT activator.

2) It would be nice to examine biochemical interactions between Dkk2 function with Lrp5/6, Ror, Krm, during neural crest development.

3) The authors present a model where Dkk2 and Wnt8 appear to function in parallel pathways that converge on β-catenin. What's not clear is why eliminating Dkk2 prevents Wnt8 from its normal activation of β-catenin and neural crest cell formation and also why vice versa, Dkk2 can't activate β-catenin and specify neural crest cells in the absence of Wnt8. Do Dkk2 and Wnt8 somehow impact each other's function?

4) Better quantification would be appropriate.

5) On BIO experiments, it is puzzling that the Dkk1+BIO does not display expected anterior expansion of Snai2 on the control side, unlike the Dkk2+Bio…?

6) This statement in the Discussion is debatable:

"While overexpression studies have previously shown that Dkk2 can function as an activator of Wnt/β-catenin signaling when co-expressed with Fzd8 or Lrp6 (Wu et al., 2000; Brott and Sokol, 2002; Li et al., 2002), this is the first study that provides evidence for its positive role in a Wnt/β-catenin regulated developmental process."

7) It should be noted that the expanded domain of Sox2 does not correlate with neural crest cell generation. This contrasts with recent studies in *Xenopus* arguing for such a relationship and maintenance of pluripotency in neural crest cells.

8) Given the proposed role of Dkk2 one would expect that it could rescue the overexpression of GSK3 to complement the BIO- inhibitor data. Has this been tested?

---

## [Author Response]

Essential revisions:1) It is critical to provide better expression profile of Dkk2 and it is suggested to move the images to main figures. The published expression pattern for Dkk2 does not show expression in the neural fold or neural crest. Furthermore the RNA seq data available on Xenbase shows a very low expression of Dkk2 during neural crest induction. It is even more striking when the expression data is compared to the Wnt partner namely Wnt8 and Lrp6. It is only in the Discussion that the authors refer to expression data presented in the supplemental material.

Although the Xenbase RNA-seq data indicates that Wnt8 and Lrp6 are expressed at higher levels than Dkk2 it is difficult to correlate this information to the amount of protein present at the time of neural crest induction. As suggested we have moved the *dkk2* expression data as a main figure of the manuscript (see Figure 4). Dkk2 is broadly expressed during embryogenesis and is not distinctly enriched in the neural crest territory. It is likely that the regionalized expression of other components of the pathway, such as Fzd receptors and Wnt ligands, which are expressed at higher levels, provides the spatiotemporal cues to achieve a localized response. The expression pattern of *dkk2* is presented and discussed in the Results section of the manuscript (see subsection “Developmental expression of Dkk2”).

2) The expression data suggest that Dkk2 is an extremely potent activator of the pathway that despite its minute expression it is essential for proper Wnt signaling. This is not entirely consistent with the fact that loss of Dkk2 can be rescued by overexpression of Wnt8, Lrp6 or β-catenin since all of these mRNA are expressed at 10 to 100 folds excess when compared to Dkk2. Does the Dkk2 protein possess some particular property such as extreme stability or extreme binding affinity to the co-receptors? Following a tagged Dkk2 protein in the embryo (by blot and IF) could provide some clues. Dose response in the embryo or animal cap could be used to determine the effectiveness of Dkk2 compare to Dkk1. What is the minimal dose at which Dkk2 expression can expend the neural crest?

Regarding the mRNA expression levels of Dkk2, Wnt8 and Lrp6 see response to point #1. As suggested we have performed Western blot analyses to evaluate the accumulation overtime of a Flag-tagged version of HuDkk2 (see Author response image 1, red arrow). Protein detection was only possible after injection of 5 ng of plasmid DNA at the 4-cell stage consistent with a previous study (Brott and Sokol, 2002). We observed that Dkk2 starts to accumulate at the gastrula stage (stages 10/12) to reach a maximum at neural stage (stages 14/18), and then progressively decreases (stage 20). A similar pattern was observed for a Flag-tagged version of HuDkk1 (not shown). Upon overexpression in the embryo, Dkk2 protein does not appear to have an unusual stability and/or degradation profile. For comparison the minimal dose at which Dkk2 can expand the neural crest in the embryo is 25 pg of plasmid DNA.

3) While the results strongly suggest that Dkk2 potentiates WNT signaling towards neural crest formation, the mechanism of action is unresolved. The authors should assess whether Dkk2 triggers expression of known direct targets of WNT/β-cat pathway, or non-canonical WNT pathways. They should also test mesoderm markers to rule out possible secondary effects.

We have performed the suggested experiment and found that with the exception of *snai2, dkk2* overexpression did not significantly upregulate other known direct targets of Wnt/β-catenin signaling such as *pax3, znf703* or *kremen2*. This result suggests that *dkk2* alone is a poor activator of Wnt/β-catenin signaling, consistent with the view that it is acting in concert with a Wnt ligand to activate the pathway. The expression of the mesoderm marker myoD was unaffected in *dkk2*-depleted embryos ruling out any possible secondary effects (see Figure 2 and Results subsection “Dkk2 is required for neural crest formation”, second paragraph).

4) The authors should test a broader repertoire of neural crest markers (in addition to Sox10 and Snai2). In addition, analysis of early stages (st. 13) is required to test effects on early NC specification markers.

We have performed additional experiments to increase the repertoire of neural crest genes analyzed, which now include *pax3, sox8, snai1, twist1* and *sox9* (see Figure 2 and Results subsection “Dkk2 is required for neural crest formation”, second paragraph). We found that early neural crest markers (neural border specifiers) such as *pax3, sox8* and *snai1* were only mildly affected – their expression levels were unchanged but their expression domain appeared to be shifted laterally – while other markers (neural crest specifiers) such as *twist1*, and *sox9* to a lesser extent, were downregulated, consistent with the phenotype observed for *sox10* and *snai2*. This result suggests that *dkk2* may not participate in neural plate border specification but rather in the neural crest progenitors formation and/or maintenance.

5) In the absence of lineage tracing, the loss of snai2 and sox10 expression is not conclusive evidence for loss of neural crest cells. The authors should examine non-neural ectoderm markers. They should also let embryos grow to later stages, as loss of neural crest should result in complete unilateral absence of craniofacial cartilage and cranial peripheral nervous system. If so, is this rescued by β-catenin and Lrp6 injections in Dkk2-depleted embryos?

We have analyzed the expression of keratin, a gene expressed in the non-neural ectoderm, and show that its overall expression is reduced/shifted laterally consistent with the expansion of *sox2* expression domain (see Figure 2B and Results subsection “Dkk2 is required for neural crest formation”, second paragraph).

We also show that neural crest-derived craniofacial cartilages in addition to pigment cells are affected upon Dkk2 depletion (see Figure 1K and Results subsection “Dkk2 is required for neural crest formation”, end of first paragraph), pointing to a loss of neural crest lineages in these embryos.

6) While the loss of function phenotype is compelling, the in vitro translation does not allow one to test the effective amount to inject in an embryo. Blocking the translation of a tagged Dkk2 mRNA would help define that dose. In addition, rescuing Dkk2 MO with an insensitive mRNA would confirm the specificity of the phenotype. The splice morpholino need to be more carefully quantified. Using realtime qPCR what is the percentage of properly spliced mRNA with various doses of the MO?

In this study, we used two MOs interfering with Dkk2 mRNA translation and splicing, respectively. Both gave a similar phenotype resulting in the inhibition of *snai2* and *sox10* expression, associated with an expansion of *sox2* expression domain. This phenotype was observed in both whole embryos and animal cap explants. We also used a control MO, which did not affect the expression of these genes. Importantly, this phenotype was efficiently rescued by expression of Lrp6 and β-catenin, two components of Wnt/β-catenin signaling pathway. We believe that this combination of approaches makes a compelling case for a specific requirement of *dkk2* in neural crest formation. It is difficult to estimate the efficacy of a translation blocking MO without a good antibody, and we have not been able to identify one that could detect endogenous *dkk2* in *Xenopus*. As suggested, we have performed qRT-PCR to evaluate the proportion of spliced versus unspliced transcripts generated upon injection of increasing doses of MO. These data are presented in Figure 1—figure supplement 1.

Other suggested revisions:1) The authors should references previous studies that provide molecular insights into how Dkk2 activates Wnt/β-cat. For example, Li et al. (2002) suggested that it is the C2 domain of Dkk2 that is critical in its WNT activating functions, and further provide evidence suggesting that Dkk2 WNT inductive role operates through DVL and GSK3 independent mechanisms. Mao and Niehrs (2003) suggested that Krm2 is responsible for the WNT-antagonistic effect of Dkk2, and that in its absence, Dkk2 operates as a WNT activator.

Both studies are now referenced in the text (see Discussion, end of second paragraph and end of fourth paragraph).

2) It would be nice to examine biochemical interactions between Dkk2 function with Lrp5/6, Ror, Krm, during neural crest development.

We agree with the reviewer, and this is something we are planning to explore in order to elucidate the precise mechanism of Dkk2-mediated signaling during neural crest specification, however we feel that this is beyond the scope of the current study.

3) The authors present a model where Dkk2 and Wnt8 appear to function in parallel pathways that converge on β-catenin. What's not clear is why eliminating Dkk2 prevents Wnt8 from its normal activation of β-catenin and neural crest cell formation and also why vice versa, Dkk2 can't activate β-catenin and specify neural crest cells in the absence of Wnt8. Do Dkk2 and Wnt8 somehow impact each other's function?

Our model predicts that signaling by Dkk2 and Wnt8 are both independently required to induce neural crest cells, this is specifically supported by the BIO experiment. Further work will address the mechanisms underlying this dual requirement during neural crest formation.

4) Better quantification would be appropriate.

For all injection experiments we provide a rigorous quantification of the phenotypes, which are summarized in the form of graphs for all figures, supplemented with numerical data provided as “source data” files.

5) On BIO experiments, it is puzzling that the Dkk1+BIO does not display expected anterior expansion of Snai2 on the control side, unlike the Dkk2+Bio?

We have included a more representative image for Dkk1+BIO (see Figure 8).

6) This statement in the Discussion is debatable:"While overexpression studies have previously shown that Dkk2 can function as an activator of Wnt/β-catenin signaling when co-expressed with Fzd8 or Lrp6 (Wu et al., 2000; Brott and Sokol, 2002; Li et al., 2002), this is the first study that provides evidence for its positive role in a Wnt/β-catenin regulated developmental process."

We have revised this statement (see Discussion, first paragraph).

7) It should be noted that the expanded domain of Sox2 does not correlate with neural crest cell generation. This contrasts with recent studies in Xenopus arguing for such a relationship and maintenance of pluripotency in neural crest cells.

We agree with the reviewer, *sox2* expression does not behave in a way that is compatible with the model of pluripotency retention in the neural crest lineage – a model that is still up for debate. At the neurula stage, we consider *sox2* as a marker for neural progenitors and not as much an indicator of pluripotency.

8) Given the proposed role of Dkk2 one would expect that it could rescue the overexpression of GSK3 to complement the BIO- inhibitor data. Has this been tested?

Because our model predicts that signaling by *dkk2* and *wnt8* are both independently necessary to induce neural crest we did not expect that overexpressing *dkk2* would be sufficient to restore neural crest formation in embryos injected with GSK3. We have performed the suggested experiment and found that Dkk2 expression was indeed unable to rescue neural crest formation in embryo overexpressing GSK3 (see Author response image 2).

**Author response image 2. respfig2:**